# Transcriptome Analyses Provide Novel Insights into Heat Stress Responses in Chieh-Qua (*Benincasa hispida* Cogn. var. Chieh-Qua How)

**DOI:** 10.3390/ijms20040883

**Published:** 2019-02-18

**Authors:** Min Wang, Biao Jiang, Wenrui Liu, Yu’e Lin, Zhaojun Liang, Xiaoming He, Qingwu Peng

**Affiliations:** 1Vegetable Research Institute, Guangdong Academy of Agricultural Sciences, Guangzhou 510640, China; wangmin1989sun@163.com (M.W.); jiangbiao@gdaas.cn (B.J.); liuwenrui@gdaas.cn (W.L.); cucumber200@163.com (Y.L.); liangzhaojun@gdaas.cn (Z.L.); 2Guangdong Key Laboratory for New Technology Research of Vegetables, Guangzhou 510640, China

**Keywords:** RNA-Seq, DEGs, HSPs, cytochrome P450, bHLHs, chieh-qua

## Abstract

Temperature rising caused by global warming has imposed significant negative effects on crop qualities and yields. To get the well-known molecular mechanism upon the higher temperature, we carefully analyzed the RNA sequencing-based transcriptomic responses of two contrasting chieh-qua genotypes: A39 (heat-tolerant) and H5 (heat-sensitive). In this study, twelve cDNA libraries generated from A39 and H5 were performed with a transcriptome assay under normal and heat stress conditions, respectively. A total of 8705 differentially expressed genes (DEGs) were detected under normal conditions (3676 up-regulated and 5029 down-regulated) and 1505 genes under heat stress (914 up-regulated and 591 down-regulated), respectively. A significant positive correlation between RNA-Seq data and qRT-PCR results was identified. DEGs related to heat shock proteins (HSPs), ubiquitin-protein ligase, transcriptional factors, and pentatricopeptide repeat-containing proteins were significantly changed after heat stress. Several genes, which encoded HSPs (*CL2311.Contig3* and *CL6612.Contig2*), cytochrome P450 (*CL4517.Contig4* and *CL683.Contig7*), and bHLH TFs (*CL914.Contig2* and *CL8321.Contig1*) were specifically induced after four days of heat stress. DEGs detected in our study between these two contrasting cultivars would provide a novel basis for isolating useful candidate genes of heat stress responses in chieh-qua.

## 1. Introduction

Heat stress due to high ambient temperatures caused by global warming has posed a serious threat to crop production worldwide and most likely the damage level will remain on an upward trend in future [1,2]. Heat stress impacts seed germination, photosynthesis, and respiration action in plants [3]. And when exposed to heat stress, plants often generated reactive oxygen species (ROS), antioxidant substances, plant hormones and other secondary metabolites [4]. In addition, continuous rising temperature leads to the change in geographical distribution and growing seasons of plants [5].

To survive in the ambient temperature conditions, plants have to evolve a multiple of internal tolerant strategies, such as the protection of heat shock proteins (HSPs), the level changes of phytohormones, and the scavenging of reactive oxygen species (ROS) by different oxidation- reduction enzymes [6,7,8]. As molecular chaperones, HSPs play crucial roles in facilitating native protein folding and preventing irreversible denatured protein aggregation, which could help plants enhance heat resistance [9,10]. In peanuts, accumulation of small HSPs could improve its resistance to heat stress [11]. Phytohormones signal transduction or content such as abscisic acid (ABA), salicylic acid (SA), and ethylene (ETH) are reported to be induced by heat stress in different plant species [12,13]. For example, ABA content levels are significantly induced when exposed to higher temperatures in pea [14,15]. Studies report that elevated levels of antioxidants are associated with plant increased thermotolerance [16]. Heat stress could induce the increase of dehydrogenase (NDH) content, which protected plants against photo-oxidation [17].

Thermotolerance studies on vegetables mainly focused on the HSPs and related enzymes. *CaHSP24* in pepper is significantly induced by high temperature, leading to its thermotolerance [18]. Ectopic expression of *BADH* of spinach in tobacco improves the glycine betaine content together with its increased resistance to heat stress [19]. Furthermore, under high temperature stress, ABA could induce the expression of *HSP70*, leading to the increased heat tolerance in cucumber [20].

Transcriptome sequencing of various resistant species is becoming a suitable technique for exploring related resistant genes and searching various biological pathways involved in the different stresses [21,22]. RNA-sequencing has been widely employed in performing crucial agricultural traits such as fruit development, parthenocarpy, flower sex expression, and other plant response to abiotic stresses in cucumber [23], watermelon [24], pumpkin [25], and other Cucurbitaceae vegetables [26]. Chieh-qua (*Benincasa hispida* Cogn. var. Chieh-qua How) is an important vegetable in Cucurbits, which belongs to a subspecies of wax gourd (*Benincasa hispida* (Thunb.) Cogn.) and widely distributed all over the South China and Southeast Asian countries [27]. High temperature caused serious loss of quality and production during the chieh-qua growth especially in the open field cultivation. However, little knowledge about the molecular studies on chieh-qua of heat stress is uncovered.

In this study, we obtained two differently heat resistant chieh-qua cultivas and firstly carried out RNA-sequencing analysis to explore the transcriptional variations under normal and high temperature conditions, respectively. Different stress-responsive novel transcript isoforms were identified and genes related to heat shock proteins (HSPs), ubiquitin-protein ligase, transcriptional factors, and pentatricopeptide repeat-containing proteins were significantly changed after heat stress. Functional categorization of differentially expressed transcripts was carried out to reveal various metabolic pathways involved in heat stress responses. This study provides a theoretical basis in the regulatory mechanism on heat tolerance in chieh-qua.

## 2. Results

### 2.1. Phenotypes of A39 and H5 under Normal and Heat Conditions

Seedlings with two true leaves of A39 and H5 (Figure 1A) were treated with a high temperature for five days and recovered for three days (Figure 1B). Both A39 and H5 showed vigorous development before heat stress. However, when treated by high temperature, A39 began to exert wilting at the top of the growth point, and its leaves turned chlorotic and yellow (Figure 1B). Using the scanning electron microscopy (SEM), we found that the number of stomas in A39 (Figure 1C) was much less than H5 (Figure 1D) in the same field size, indicating that H5 lost water more easily when encountered to heat stress. The relative conductivity of H5 was significantly higher than A39 (Figure 1E), indicating the membrane lipid showed much higher degree of damage in H5. Approximately 30% of the heat treated H5 plants survived after the subsequent three days recovery, compared with 70% of A39 plants (Figure 1F). There was no difference of enzyme superoxide dismutase (SOD) between A39 and H5 before heat, while H5 presented prominent decrease of SOD at the fourth day after heat stress (Figure 1G).

### 2.2. Transcripts Assembly of A39 and H5

In order to explore the transcriptional variations between A39 and H5 under normal and heat condition, respectively, the RNA-sequencing assay was carried out. A total of about 45–55 million clean reads were obtained per sample (Table 1) after removing low-quality and adaptor-containing raw reads. Finally, we identified 8705 (Appendix A) and 1505 (Appendix A) differentially expressed genes (DEGs) in the comparison of A39 versus H5 under normal conditions and heat stress, respectively. Among them, under normal conditions, 3676 genes were up-regulated and 5029 down-regulated (gene expression in H5 compared with that in A39) (Figure 2A). Additionally, 914 up-regulated and 591 down-regulated genes were identified at the fourth day after heat treatment (Figure 2B). All the original data have been uploaded in the NCBI SRA data (SRP180414). Furthermore, in order to identify the accuracy of the RNA-sequencing data, we randomly selected 16 DEGs and performed a qRT-PCR assay. The data demonstrated that there existed a strong positive correlation (two tailed, R^2^ = 0.938) between the RNA-seq data and qRT-PCR results (Figure 3), indicating the accuracy of the RNA-Seq results.

### 2.3. Functional Classification of Heat Response Genes

The GO (gene ontology) standardized classification system for gene function was employed to analyze DEGs and understand the molecular events involved in the heat response. Three categories including “biological process (BP)”, “molecular function (MF)”, and “cellular components (CC)” were classified under normal conditions (Appendix A) (Appendix A) and heat stress (Appendix A) (Appendix A), respectively. Under normal conditions, the largest number of genes were mainly in the CC category (integral component of membrane), while the main category was changed to MF (ATP binding) at the four days after heat treatment, followed by the RNA binding and zinc ion binding metabolic process (Appendix A).

In order to examine the DEG-associated pathways, they were searched using the KEGG pathway database. The top 20 enriched pathways were shown in Figure 4. Main pathways under normal condition were “biosynthesis of secondary metabolites”, “plant hormone signal transduction”, and “purine metabolism” (Figure 4A) (Appendix A). Especially, for the general metabolism pathways, DEGs were mostly enriched including the starch and sucrose, pyrimidine, purine, and galactose under control (Figure 4A). When exposed to heat stress, genes related to “splicesome”, “plant hormone signal transduction”, “purine metabolism” and “protein processing in endoplasmic reticulum” were mostly enriched (Figure 4B) (Appendix A), indicating these pathways and processes possibly participated in seedling heat resistance.

### 2.4. Expression of Genes Encoding HSPs, HSFs, and Cytochrome P450

Based on the GO and KEGG analysis, several DEGs encoding heat shock proteins and ubiquitin-protein ligase were selected to perform qRT-PCR assay. A total of 6 transcripts were chosen, including four genes with heat shock proteins (HSPs) or heat stress transcription factor (HSFs) and two genes with response to water deprivation (Table 2). The qRT-PCR experiment was carried out to validate RNA-seq results under normal and heat treatment, respectively. These results showed that during heat stress, all these genes expression were induced and genes in H5 were significantly down-regulated compared with those in A39 (Figure 5). In addition, we typically selected six genes, which encoded ubiquitin-protein ligase to detect their expression changes (Table 3). The above results showed that the expression of all the six genes was observably decreased in H5 during heat stress in contrast to those in A39 (Figure 6). All these results of qRT-PCR were consistent with the RNA-sequencing data.

### 2.5. Expression Analysis of Genes Related to TFs and Pentatricopeptide Repeat-Containing Protein

In this study, five TFs were selected and performed the qRT-PCR assay to detect their expression under control and heat stress, respectively. We found that two bHLHs (bHLH128 and bHLH143), two PIF1-like, and one PCL1-like TF were significantly down-regulated in H5 under heat stress in comparison with those in A39 (Figure 7) (Table 4). Under normal conditions, three TFs showed no differences between A39 and H5, and the expression of other two TFs (bHLH143 and PIF1-like) were decreased in H5. In addition, the expression of four PPR proteins changed significantly after heat treatment (Figure 7) (Table 4). The above data demonstrated that their transcription were all prominently down-regulated in H5 compared with A39, although the differences of expression existed before treatment.

## 3. Discussion

Previous studies demonstrated that the analysis and availability of diverse genetic resources could provide crucial information for understanding the molecular basis of variability in their response to abiotic stresses [21]. In the present study, we characterized and analyzed two chieh-qua genotypes (A39 and H5) for their significantly different responses to heat stress. H5 showed heat sensibility in a higher temperature condition with increased relative conductivity and decreased survival rate, as well as SOD enzyme activity. Combing the analysis of RNA-seq, multiple of DEGs were up-regulated and down-regulated significantly under heat stress. Among them, several DEGs related to heat shock proteins, ubiquitin-protein ligase, and some TFs were possibly involved in the heat response tolerance with prominent expression changes between the two chieh-qua materials.

### 3.1. H5 Has More Stomas in the Leaf Than A39

The regulation of stomatal opening and closure is crucial to the normal transpiration of water loss and plays crucial roles in the stress resistances [28]. Rice *am1* mutant have high percentage of completely closed stomata, leading to the increased drought resistance compared with the wild type [29]. *Dca1* mutant with drought tolerance has less number of stomata than the control [30]. Although stomata are mostly reported in the drought resistance studies, here, we found that the number of stomas in A39 was less than that in H5 in the same field size, indicating that the global differences in morphogenetic response between A39 and H5 might serve as a reason of different heat resistance.

### 3.2. Analysis of HSPs, HSFs During Heat Stress

HSPs comprising several evolutionarily conserved protein families, exert crucial roles in heat stress and could be immediately induced by high temperatures [13]. In *Arabidopsis*, HSP101 plays a pivotal role in heat tolerance [31]. Overexpressing of a small heat shock protein, sHSP17.7, increases the heat tolerance of transgenic rice [32]. The A1 class of heat shock factors (HSFA1s) is crucial to heat and other stresses in *Arabidopsis* [33]. For example, HSFA3 functions in both heat stress and water-deficit stress responses in *Arabidopsis* [34]. In our study, we found that the expression of two respective HSPs and HSFs were significantly decreased in H5 in comparison with A39 at the fourth day of heat stress, which indicated that high expression of these genes in A39 might contribute to its tolerance to heat stress to some extent.

### 3.3. Analysis of Ubiquitin-Protein Ligase During Heat Stress

The ubiquitin/26S proteasome system (UPS) is a critical regulatory mechanism that controls plants response to biotic and abiotic stresses [35,36]. The UPS consists of three key enzymes, ubiquitin activating enzymes (E1), ubiquitin-conjugating enzymes (E2), and ubiquitin ligases (E3) [37,38]. *AtCHIP*, encoding a u-box-containing E3 ubiquitin ligase, is up-regulated by several stresses such as the high temperature in *Arabidopsis* [39]. Overexpressing the RING E3 ligase gene *SDIR1* enhances drought tolerance by positively regulating the ABA signaling pathway in *Arabidopsis* [40]. Our study detected six ubiquitin-protein ligase genes and the expression of them was significantly decreased in H5 when compared with A39 after four days of heat stress, especially the *CL1411.Contig3*, *CL617.Contig22*, and *CL10863.Contig1*, suggesting that low expression might be the reason for the sensitivity of H5 to high temperature.

### 3.4. Analysis of TFs During Heat Stress

Previous studies had reported that most types of transcription factors (TFs), such as bHLH and phytochrome-interacting factors (PIF) exerted crucial roles when plants exposed to abiotic stresses such as drought, cold, salt stress, and others mostly by mediating anthocyanin biosynthesis and plant hormone signaling [41,42]. Several studies in rice reported that bHLH played important roles in response to different stresses, for example *OsbHLH1* in cold response [43], *RERJ1* in drought response [44], *OsPTF1* in tolerance to phosphate starvation [45], and *OrbHLH2* in tolerance to salt and osmotic stress [46]. In our present study, we also found the significant expression changes of these TFs between A39 and H5 during heat stress, indicating the chieh-qua bHLHs were involved in the resistance on high temperatures.

Phytochrome-interacting factors (PIFs), as a subfamily of basic helix-loop-helix (bHLH) transcription factors, played important roles in regulating plant responses to stresses. Ectopic over expression of maize *ZmPIF3* improved the resistance on drought and salt in rice [47,48]. Overexpression PIF improved drought stress tolerance in transgenic rice. Two PIFs (*CL6744.Contig4* and *CL6624.Contig3*) were prominently decreased in H5 in contrast to A39, implying high expression of PIFs in A39 might increase the resistance on heat stress.

### 3.5. Analysis of PPRs During Heat Stress

Pentatricopeptide repeat-containing proteins (PPRs), defined by a tandem array of a PPR motif consisting of 35 amino acids, were most involved in regulating RNA splicing, editing, cleavage, stability, and translation during plant development and growth [49,50]. In addition, PPRs also participated in the plant stress resistance. For example, *WSL5* (pentatricopeptide repeat protein) is essential for plant cold tolerance by regulating chloroplast biogenesis [51]. In our study, four PPRs were significantly down-regulated in H5 when compared with A39 at the four days after heat stress, indicating high expression of PPRs might contribute to A39′s resistance to high temperature.

### 3.6. Differences of Genes Expression under Control

Under control conditions, multiple DEGs were detected, which were mostly involved in the metabolism pathways such as the starch and sucrose, pyrimidine, purine, and galactose. Take the starch and sucrose metabolism for example, previous studies reported that this metabolism pathway played important role in the responses to various stresses such as salt [52], water [53,54], and drought [55]. Our study identified that the chieh-qua cultivars with heat-resistant differences exerted significantly differential expression on starch and sucrose metabolism, indicating this DEGs might contribute to their resistant differences on heat when exposed to high temperature condition.

In all, the RNA-Seq between different chieh-qua cultivars was firstly carried out to analyze the regulatory mechanism under high temperature. Several crucial genes related to heat shock proteins, ubiquitin-protein ligase, and transcriptional factors were significantly important during heat stress. This work not only provided a basis for further understanding of the molecular mechanism on heat tolerance, but also exerted valuable and useful genes involved in heat stress, which would be helpful for the genetic improvement of heat tolerance in chieh-qua breeding. Further research should focus on the functional characterization of crucial genes involved in heat response combing physiology, genetics, and molecular biology technology. Most importantly, crucial genes’ roles on heat stress should be identified by transgenic study and other methods. 

## 4. Materials and Methods

### 4.1. Plant Materials and Heat Stress Treatment

A39 (heat-tolerant) was a homozygous inbred line derived from the native variety “Qixingjie” in the Guangdong province. H5 (heat-sensitive) was also a native variety of the Guangdong province. Seeds were soaked in warm water for 6–8 h in room temperature and germinated 2 days on a wet filter in a culture dish at 30–32 °C under a dark environment. Subsequently, these germinated seeds of A39 and H5 were grown in a feeding block under 14 h/10 h with 30 °C/24 °C in day/night, respectively, in a culture room (5500 lux, 60% RH). When plants were grown to two true leaves stage, they were transferred to a high temperature environment for five days (45 °C/40 °C in day/night). After that, seedlings were recovered for 3 days to normal temperature. There were a total of 120 plants for A39 and H5 seedlings, respectively. Before treatment and on the 4th day of heat stress, leaf tissue of 10 plants from each pot was sampled and pooled together from normal plants and heat stressed plants of each cultivar, respectively. Three biological replicates were applied for each cultivar. Each biological replicate contained 10 plants randomly. These samples were immediately frozen in liquid nitrogen and consistently stored at −80 °C for further analysis. Scanning electron microscopy (SEM).

Leaves of A39 and H5 under normal conditions were air-dried. Leaf abaxial epidermis was visualized under a HITACHI SU8020 variable pressure SEM (Hitachi, Tokyo, Japan) and imaged using an H-7500 transmission electron microscope (Hitachi, Japan). In addition, in view of changes along one simple leaf, we used the same leaf part and the leaves of the same sizes for management and investigation.

### 4.2. Measurement of SOD Activity

SOD activity of A39 and H5 was detected by the NBT method (nitroblue tetrazolium) [56,57]. In detail, the chieh-qua leaves (1 g) were homogenized in liquid nitrogen and mortar in 10 mL of 0.2 M lysate (citrate phosphate buffer (pH 6.5) with 0.5% Triton X-100). Then, the mixture was centrifuged at 10,000× *g* for 30 min at 4 °C and the supernatant was the enzyme source. The riboflavin was added last after the enzyme extract was finished and A560 was measured for their OD values.

### 4.3. Transcriptome Sequencing

A total of twelve samples (three biological replicates each of A39 and H5 at normal and heat stress, respectively) were used for RNA extraction with the TRIZOL reagent according to protocol (TaKaRa, Japan). RNA was then purified (using DNAse) and concentrated using an RNeasyMinElute clean up kit (Qiagen, Germany). Then, cDNA libraries were constructed from RNA samples for Illumina paired-end (PE) sequencing following the Illumina protocol. PE sequencing (2 × 100 bp) was performed on the IlluminaHiSeq 2000 platform (Illumina, San Diego CA, USA) at Novogene Bioinformatics Technology Co., Ltd. (Beijing, China).

### 4.4. Screening and Significant Test for Differentially Expressed Genes (DEGs)

Gene expression level was calculated by quantifying reads according to RPKM (reads per kilobase per million reads) method [58]. Then the NOISeq was used to identify DEGs, which existed in normal and heat stress transcriptome libraries according to the following criteria: Fold change ≥ 2 and divergence probability ≥ 0.8. GO enrichment for these DEGs was performed using WEGO software [59]. To further understand the DEG biological functions, pathway enrichment analysis was carried out on the basis of KEGG database [60].

### 4.5. Quantitative Real-Time PCR Analysis

Total RNA was extracted using TRIZOL Kit (TaKaRa, Japan). First-strand cDNA was obtained from 2 μg RNA combing the QuantiTect Reverse Transcription Kit (Qiagen, Germany). qRT-PCR (20 μL volume) was performed with 0.5 μL of cDNA, 0.2 μM of primer mix, and the SYBR Premix Ex Taq Kit (TaKaRa, Japan). The endogenous chieh-qua *CqActin* gene (F: TCAACCCAAAGGCTAACAG; R: CTTGTCCATCAGGCAGTTC) was used as the reference genes. The experiment was carried out with three technical and biological replicates, respectively. All qRT-PCR primers are listed in Appendix A.

### 4.6. Statistical Analysis

Heat-map and cluster analysis were carried out by R-3.4.2 and MEGA6 software. Significant differences were detected by IBM SPSS Statistics 20. Gene relative expressions were calculated using the 2^−∆∆*C*t^ method [61]. In addition, GraphPad Prism5 was used for chart preparation.

## Figures and Tables

**Figure 1 ijms-20-00883-f001:**
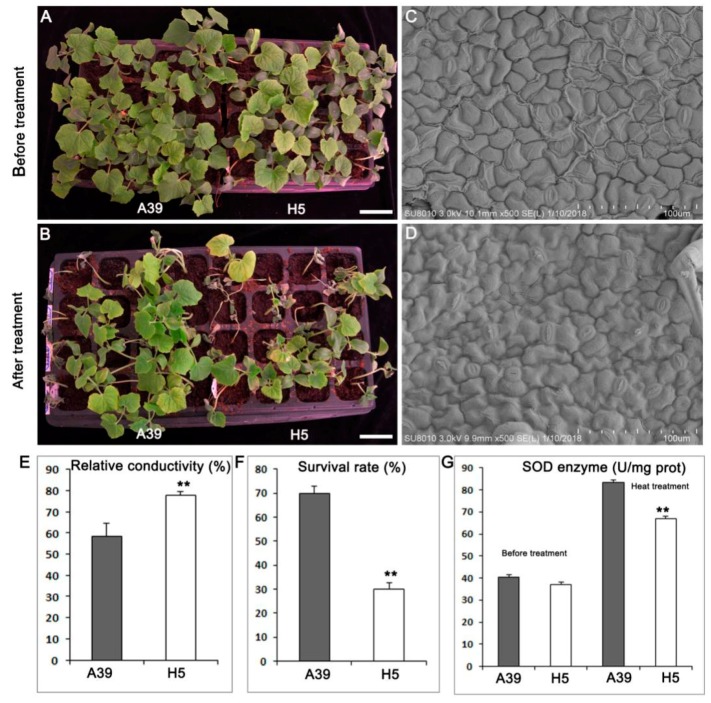
Phenotypes of A39 and H5 after four days heat stress. (**A**) A39 and H5 plants were grown under normal condition for 14 days. (**B**) Seedlings were watered and grew under normal conditions for three days. (**C**,**D**) Scanning electron microscopy (SEM) images of leaves in A39 (**C**) and H5 (**D**). (**E**) Detection of relative conductivity of A39 and H5. (**F**) The survival rate of plants of A39 and H5 following the four days heat treatment. (**G**) Measurement of superoxide dismutase (SOD) enzyme activity under normal condition and four days after heat stress. Data are presented as the mean ± standard deviation (*n* = 9). ** *p* < 0.01; Student’s *t*-test.

**Figure 2 ijms-20-00883-f002:**
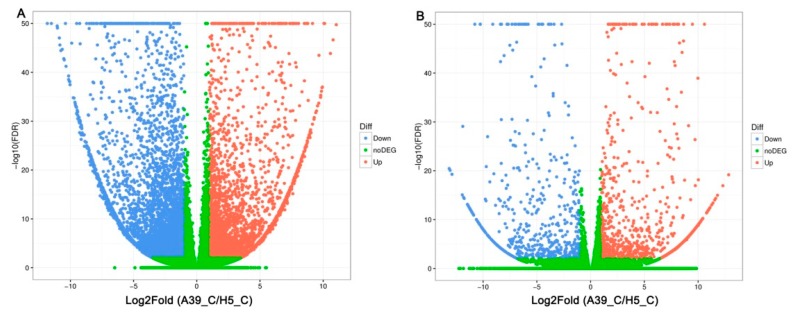
Comparison of different genes expression (DEGs) in leaves between A39 and H5 under normal conditions (**A**) and four days after heat stress (**B**). X- and y-axes represent log2 values of gene expression. Red, green, and blue correspond to up-regulated, unaltered, and down-regulated gene expression, respectively.

**Figure 3 ijms-20-00883-f003:**
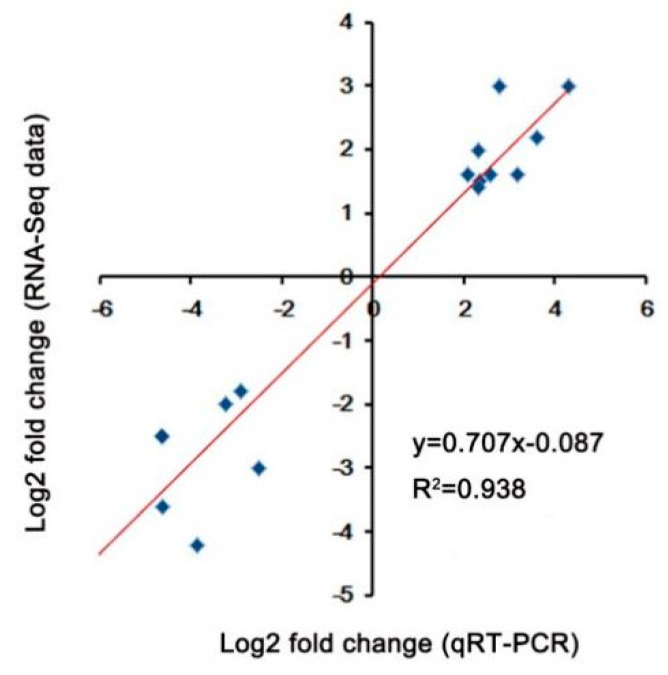
Validation of DEGs under heat stress. Correlation between the fold change analyzed by RNA-seq (*y*-axis) and data obtained using qRT-PCR (*x*-axis).

**Figure 4 ijms-20-00883-f004:**
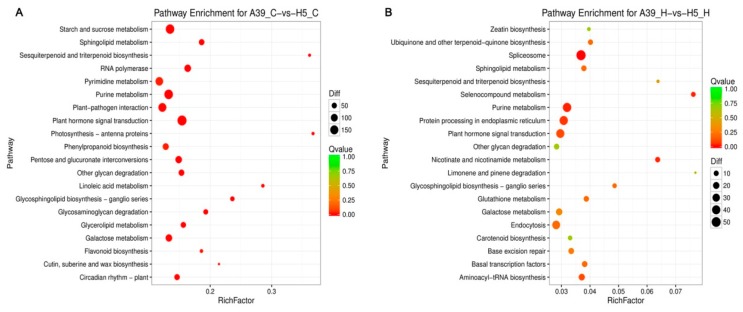
KEGG enrichments of annotated DEGs across three comparisons of normal condition (**A**) and heat stress (**B**). The *y*-axis indicates the KEGG pathway and the *x*-axis indicates the enrichment factor. A high q-value is represented by light blue, and a low q-value is represented by dark blue.

**Figure 5 ijms-20-00883-f005:**
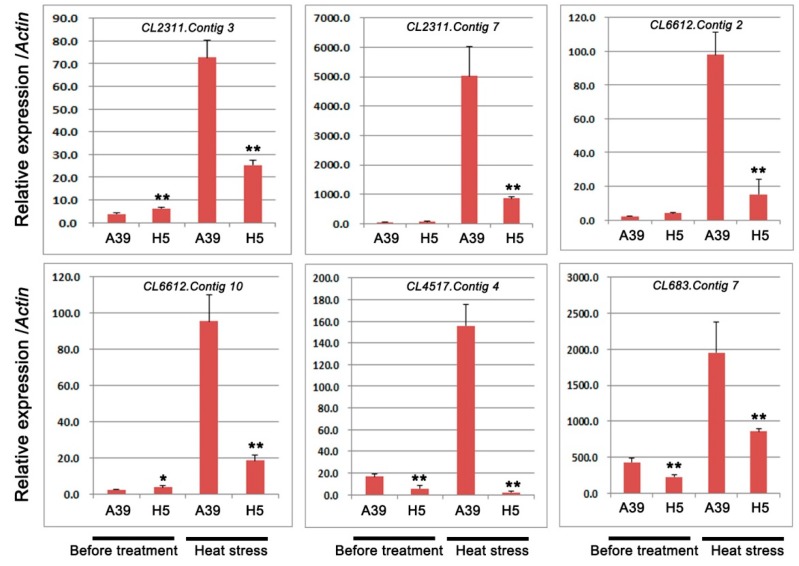
Relative expression of genes related to heat shock proteins (HSPs), heat stress transcription factor (HSFs), and cytochrome P450. Data are presented as the mean ± standard deviation (*n* = 9). * 0.01 ≤ *p* ≤ 0.05, ** *p* ≤ 0.01, Student’s *t* test.

**Figure 6 ijms-20-00883-f006:**
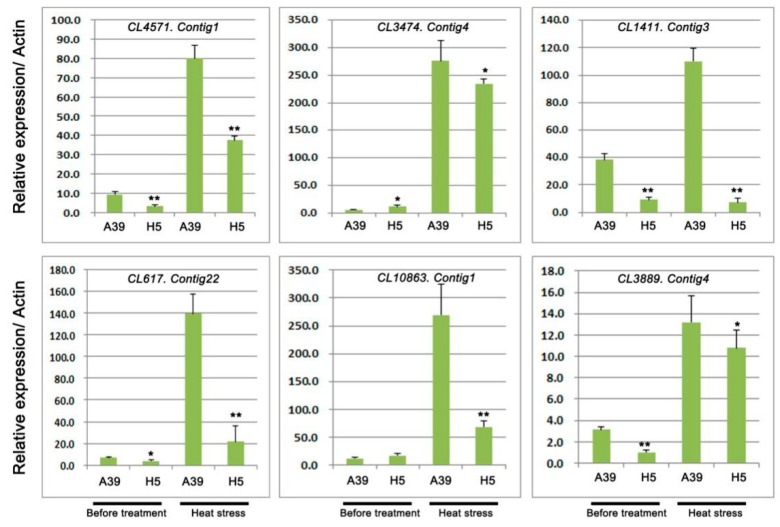
Relative expression of genes related to ubiquitin carboxyl-terminal hydrolases and E3 ubiquitin-protein ligases. Data are presented as the mean ± standard deviation (*n* = 9). * 0.01 ≤ *p* ≤ 0.05, ** *p* ≤ 0.01, Student’s *t* test.

**Figure 7 ijms-20-00883-f007:**
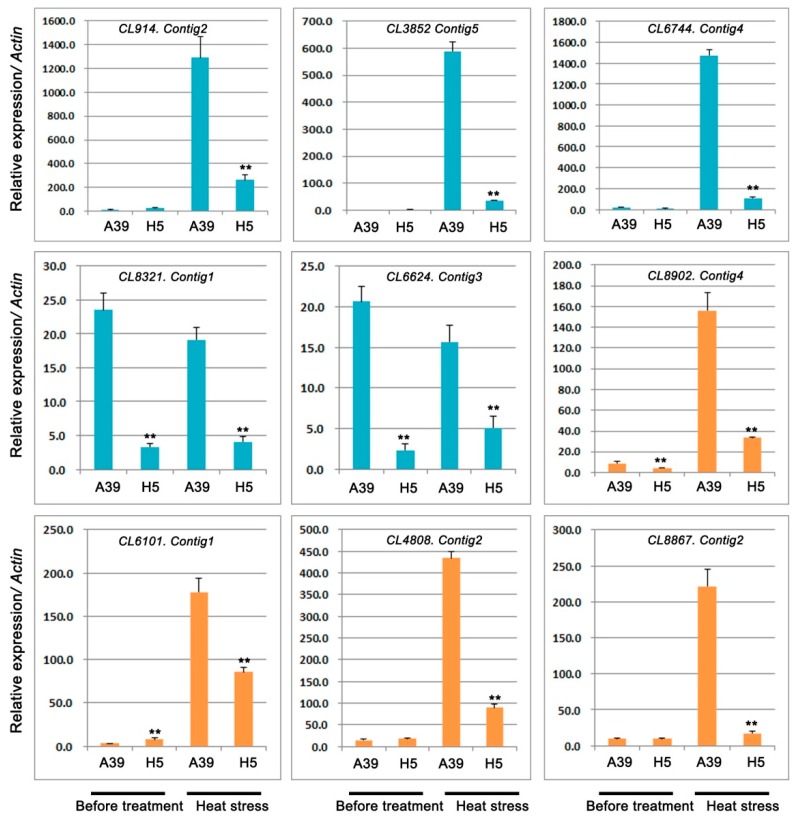
Relative expression of genes related to transcription factors (bHLHs, PCL1s, PIF1s) and pentatricopeptide repeat-containing protein. Data are presented as the mean ± standard deviation (*n* = 9). ** *p* ≤0.01, Student’s *t* test.

**Table 1 ijms-20-00883-t001:** Sample filtering statistics of A39 and H5

Type	A39_C	H5_C	A39_H	H5_H
Read Length	150	150	150	150
Total Raw Reads	45,345,693.33	54,886,327.3	47,159,128	50,066,083.33
Total Raw Bases	6,801,854,000	8,232,949,100	7,073,869,200	7,509,912,500
Total Clean Reads	45,233,028	54,676,526.67	47,003,104	49,889,418
Total Clean Reads Ratio (%)	99.75	99.63	99.67	99.65
Total Clean Bases	6,784,954,200	8,201,479,000	7,050,465,600	7,483,412,700
Total Clean Bases Ratio (%)	99.75	99.63	99.67	99.65
Total Adapter Reads	89,349.3	181,258.67	126,836.7	150,814
Total Adapter Reads Ratio (%)	0.2	0.32	0.27	0.3
Total Low-Quality Reads	23,316	28,542	29,187.3	25,851.33
Total Low-Quality Reads Ratio (%)	0.05	0.05	0.06	0.05
Clean Reads GC (%)	45.49	44.67	44.82	44.29
Clean Reads Q20 (%)	97.17	97.25	97.02	97.21
Clean Reads Q30 (%)	93.15	93.31	92.89	93.27

**Table 2 ijms-20-00883-t002:** Genes related toHSPs, HSFs, and cytochrome P450.

GeneID	log2 Fold	*p*-Value	Diff	Nr-Annotation
*CL2311.Contig3*	−10.26	9.62 × 10^−267^	Down	heat shock protein (HSPs)
*CL2311.Contig7*	−3.92	2.92 × 10^−239^	Down	heat shock protein(HSPs)
*CL6612.Contig2*	−11.46	7.01 × 10^−16^	Down	heat stress transcription factor(HSFs)
*CL6612.Contig10*	−4.79	5.23 × 10^−6^	Down	heat stress transcription factor (HSFs)
*CL4517.Contig4*	−1.02	2.8121 × 10^−4^	Down	cytochrome P450
*CL683.Contig7*	−1.10	3.19 × 10^−16^	Down	cytochrome P450

**Table 3 ijms-20-00883-t003:** Genes related to ubiquitin-protein ligase.

GeneID	log2 Fold	*p* Value	Diff	Nr-Annotation
*CL4571.Contig1*	−2.41	9.88 × 10^−8^	Down	ubiquitin carboxyl-terminal hydrolase 2
*CL3474.Contig4*	−8.74	3.91 × 10^−7^	Down	ubiquitin carboxyl-terminal hydrolase 14
*CL1411.Contig3*	−1.42	1.07 × 10^−22^	Down	ubiquitin domain-containing protein DSK2a-like
*CL617.Contig22*	−10.09	8.83 × 10^−11^	Down	E3 ubiquitin-protein ligase UPL7
*CL10863.Contig1*	−8.26	3.16 × 10^−6^	Down	E3 ubiquitin-protein ligase COP1
*CL3889.Contig4*	−8.03	9.51 × 10^−6^	Down	ubiquitin carboxyl-terminal hydrolase 22-like

**Table 4 ijms-20-00883-t004:** Genes related to transcription factors and pentatricopeptide repeat-containing protein.

GeneID	log2 Fold	*p* Value	Diff	Nr-Annotation
*CL914.Contig2*	−3.97	8.54 × 10^−5^	Down	transcription factor bHLH128
*CL3852.Contig5*	−8.00	1.09 × 10^−4^	Down	transcription factor PCL1-like
*CL6744.Contig4*	−5.72	2.52 × 10^−11^	Down	transcription factor PIF1-like
*CL8321.Contig1*	−7.73	4.47 × 10^−100^	Down	transcription factor bHLH143-like
*CL6624.Contig3*	−8.54	5.01 × 10^−17^	Down	transcription factor PIF1-like
*CL8902.Contig4*	−8.49	1.33 × 10^−5^	Down	pentatricopeptide repeat-containing protein
*CL6101.Contig1*	−7.47	6.24 × 10^−8^	Down	pentatricopeptide repeat-containing protein
*CL4808.Contig2*	−3.06	7.6 × 10^−12^	Down	pentatricopeptide repeat-containing protein
*CL8867.Contig2*	−9.98	1.95 × 10^−10^	Down	pentatricopeptide repeat-containing protein

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
