# Peer review of "Transcriptome Analyses Provide Novel Insights into Heat Stress Responses in Chieh-Qua (Benincasa hispida Cogn. var. Chieh-Qua How)"

_ijms, 2019, doi:10.3390/ijms20040883_

Round 1
Reviewer 1 Report
This study by Wang et al., is very nice and timely.
I found the manuscript presented in a clear way, all steps well documented and with a good statistical framework. I recommend it for publication with few small comments.
Authors explained the contents very effectively. However, the manuscript requires extensive English correction.
There are many spelling mistakes for example Line # 40 "native pritein folding" should be "native protein folding".
Many time there is repetition of words for example Line # 47 "plants against against photo-oxidation" should be "plants against photo-oxidation"
Other grammatical mistakes for example Line # 232 "genes related to in heat shock" should be "genes related to heat shock".
English of this manuscript should be improved with the help of a native English speaker.
Author Response
Dear reviewer
Thank you very much for handling our manuscript carefully entitled “Leaf transcriptome analysis exploring of candidate genes for heat stress responses in chieh-qua (BenincasahispidaCogn. var. Chieh-qua How)” (Manuscript ID: 442467). We have carefully considered your comments, and revised the manuscript so that all concerns were taken into account.
This study by Wang et al., is very nice and timely.
I found the manuscript presented in a clear way, all steps well documented and with a good statistical framework. I recommend it for publication with few small comments.
Authors explained the contents very effectively. However, the manuscript requires extensive English correction.
Response: Thank you very much for your evaluation and advice. We will carefully examine the whole manuscript and correct it timely.
There are many spelling mistakes for example Line # 40 "native pritein folding" should be "native protein folding".
Response: We have changed it to “native protein folding”.
Manytime there is repetition of words for example Line # 47 "plants against against photo-oxidation" should be "plants against photo-oxidation"
Response: Thank you very much for your suggestion. We have deleted one “against” and changed it to “plants against photo-oxidation”.
Other grammatical mistakes for example Line # 232 "genes related to in heat shock" should be "genes related to heat shock".
Response: We have deleted “in” and changed it to “genes related to heat shock”.
English of this manuscript should be improved with the help of a native English speaker.
Response: Thank you very much for your advice. We have sent out manuscript to a native speaker for language improved and changes could be found in the modified version.
Reviewer 2 Report
This is potentially interesting paper describe differences in gene expression in heat tolerant and heat-sanative lines. Some corrections/clarity are required in the text.
However, the main points what must be reflected in discussion, is dramatic differences in gene expression in the control (without heat-stress).
Changes in stomata number may suggest that there is a global differences in morphogenetic response between lines with differences in cell types ratio, what, in turn, may serve as a reason of heat-tolerance.
Authors should at least discussed this point and in next research focus on such differences.
Minor comments:
Lines 14-16: sentences require edition.
Line17-18:what is the differences between genes and transcripts?
Line33: respiration is enough
Lines 34-35: ROS and hormones produced during normal growth. Do you mean temporary ROS accumulation?
Lines 38-39: please, edit: what is HSP protection? Phytohormones change? It should be level of phytohormones…
Line44: please, edit: “content level significantly reached to peak”-which lenquich is this?
Line 49: what is relative enzymes?
Line 55: edit, please!
Line 79: found, not investigated.
Line 89 (B) – please, edit. ( E) – which conductivity? After stress? (G) -why 4 days? Which SOD?
Line 93: please, edit!
Line 98: what do you mean as heat stress? 4 or 5 days?
Line 173: given facts that A39 and H5 have even more differences under normal conditions, I would rather tell about general metabolism differences what allow to better adapted to heat.
Line 239: heat-tolerant
Line 240: heat-sensitive
Line 247: what is normal leaf?
Lines 251-254: it is much easy to use polysaccharide stating for leaf structure.
Line 276: please, clarify how much replications.
Figure 6 legend/layout does not fit with the text. Please, clarify TF name on it.
Similar comments to other figures.
Author Response
Dear reviewer
Thank you very much for handling our manuscript carefully entitled “Leaf transcriptome analysis exploring of candidate genes for heat stress responses in chieh-qua (BenincasahispidaCogn. var. Chieh-qua How)” (Manuscript ID: 442467). We have carefully considered your comments, and revised the manuscript so that all concerns were taken into account.
This is potentially interesting paper describe differences in gene expression in heat tolerant and heat-sanative lines. Some corrections/clarity are required in the text.
However, the main points what must be reflected in discussion, is dramatic differences in gene expression in the control (without heat-stress).
Response: Thank you much for your suggestion. According to your advice, we have added the discussion of the dramatic differences in the gene expression under the control as the followed bellowing.
Under control condition, multiple DEGs were detected, which most involved in the metabolism pathways such as the starch and sucrose, pyrimidine, purine, and galactose. Take the starch and sucrose metabolism for example, previous studies reported that this metabolism pathway played important role in the responses refulation under various stresses such as salt (Oliveira et al., 2013), water (Chang and Ryan, 1987; Liao et al., 2016), and drought (Netrphan et al., 2012). Our study identified that the chieh-qua cultivars with heat-reisitant differences exerted significantly differential expression on starch and sucrose metabolism, indicating this DEGs might contribute to their resistant differences on heat when exposed to high temperture condition.
[1] Oliveira, Helena, Costa, et al. NaCl and Phaeomoniella chlamydospora affect differently starch and; sucrose metabolism in grapevines. Acta Scientiarum-agronomy, 2013, 35(2):153-159.
[2] Liao W B, Li YY, Lu C , et al. Expression of sucrose metabolism and transport genes in cassava petiole abscission zones in response to water stress. Biologia Plantarum, 2016:1-8.
[3] Chang CW, Ryan RD. Effects of water stress on starch and sucrose metabolism in cotton leaves. Starch, 1987, 39 (3):84-87.
[4] Netrphan S, Tungngoen K, Suksangpanomrung M, et al. Differential expression of genes involved in sucrose synthesis in source and sink organs of cassava plants undergoing seasonal drought stress. Journal of Agricultural Science, (1916-9752), 2012, 4(11):171-185.
Changes in stomata number may suggest that there is a global differences in morphogenetic response between lines with differences in cell types ratio, what, in turn, may serve as a reason of heat-tolerance.
Response: Thank you very much for your advice. We have added sentences in the discussion about the differences of stomata and morphogenetic response. “We found that the number of stomas in A39 was less than that in H5 in the same field size, indicating that the global differences in morphogenetic response between A39 and H5 might serve as a reason of different heat resistance.”
Authors should at least discussed this point and in next research focus on such differences.
Response: Thank you for your suggestion. We have pointed the next research focus on such differences in the discussion part.
Therefore, next research should focus on the functional regulation of crucial genes involved in heat response combing physiology, genetics, and molecular biology technology. Especially, the gene’s role on heat stress should be identified by transgentic study.
Minor comments:
Lines 14-16: sentences require edition.
Response: We have edited the sentences to “In this study, twelve cDNA libraries generated from A39 and H5 were performed a transcriptome assay under normal and heat stress conditions, respectively.”
Line17-18: what is the differences between genes and transcripts?
Response: For the transcriptome sequencing, we firstly obtained the transcripts. One gene might include several transcripts. Therefore, in our manuscript, the1505 transcripts should be 1505 genes.
Line33: respiration is enough
Response: We have changed the sentence to “In details, heat stress impacts seed germination, photosynthesis, and respiration action in plants.”
Lines 34-35: ROS and hormones produced during normal growth. Do you mean temporary ROS accumulation?
Response: Previous studies reported that reactive oxygenspecies (ROS), which are widely generated under various stresses, have been proposed to carry out function as second messengers in ABA signaling in guardcells (Pei et al. 2000; Kohler et al.2003; Bright et al. 2006; Suzuki et al., 2006). And one species of ROS, hydrogen peroxide (H2O2) was reported that it could induce stomata closure (Zhanget al. 2001). Therefore, under high temperature, ROS might be accumulated and a study (Suzuki et al., 2006) has verified it.
[1] Suzuki N, Mittler R. 2006. Reactive oxygen species and temperature stresses: A delicate balance between signaling and destruction. PhysiologiaPlantarum126:45-51.
[2] Pei ZM, Murata Y, Benning G, Thomine S, Klusener B, Allen GJ,Grill E, Schroeder JI. 2000. Calcium channels activated byhydrogen peroxide mediate abscisic acid signalling in guardcells. Nature 406: 731-734.
[3] Kohler B, Hills A, Blatt MR. 2003. Control of guard cell ionchannels by hydrogen peroxide and abscisic acid indicatestheir action through alternate signaling pathways. PlantPhysiol. 131: 385-388.
[4] Bright J, Desikan R, Hancock JT, Weir IS, Neill SJ. 2006. ABAinducedNO generation and stomatalclosure in Arabidopsisare dependent on H2O2 synthesis. Plant J. 45: 113-122.
[5] Zhang X, Zhang L, Dong F, Gao J, Galbraith DW, Song CP. 2001.Hydrogen peroxide is involved in abscisic acid-inducedstomatal closure in Viciafaba. Plant Physiol. 126: 1438-1448.
Lines 38-39: please, edit: what is HSP protection? Phytohormones change? It should be level of phytohormones
Response: Thank you very much for your advice. HSP protection was that the HSPs, as molecular chaperones, could help other proteins refolding, intracellular distribution, and degradation, as well as in the signal transduction chains. The levels of phytohormones such as abscisic acid (ABA) and ethylene (ETH) are mostly increased under high temperature.
We have added several appropriate sentences in the manuscript. And the former sentence has been changed to “To survive in the ambinent temperture condition, plants have to evolve a multiple of internal tolerant strategies, such as heat shock prote n (HSP) could protected other proteins refolding, intracellular distribution, and degradation, the levels of phytohormones are mostly increased under high temperature, and reactive oxygen species (ROS) were scavenged by different oxidation-reduction enzymes [6-8].”
Line44: please, edit: “content level significantly reached to peak”-which lenquich is this?
Response: Thank you very much for your advice. It’s really not appropriate for this description. And we have changed it to “ABA content levels were significantly induced when exposed to higher temperature in pea”.
Line 49: what is relative enzymes?
Response: I am sorry it is “related” not “relative”.
Line 55: edit, please!
Response: We have edited this sentence to “Transcriptome sequencingof various resistant species is becoming a suitable technique for exploring related resistant genes and searching various biological pathways involved in the different stresses [21, 22]”.
Line 79: found, not investigated.
Response: We have changed it “found”.
Line 89 (B) – please, edit. (E) – which conductivity? After stress? (G) -why 4 days? Which SOD?
Response: We have changed the Line 89 (B) to “Seedlings were watered and grew under normal condition for 3 days”.“ (E, F) Relative conductivity and survival rate of plants.” has been changed to “(E) Detection of relative conductivity of A39 and H5. (F) The survival rate of plants of A39 and H5 following the 4 days heat treatment.” (G) 4 days heat stress: because when plants were exposed to high temperature for 4 days, A39 and H5 demonstrated significant phenotypic resistant differences. SOD is referred to the superoxide dismutase.
Line 93: please, edit!
Response: We have changed it to “Transcripts assembly of A39 and H5”.
Line 98: what do you mean as heat stress? 4 or 5 days?
Response: Heat stress refers to plants exposed to high temperature condition for 4 or 5 days.
Line 173: given facts that A39 and H5 have even more differences under normal conditions, I would rather tell about general metabolism differences what allow to better adapted to heat.
Response: Thank you very much for your advice. We have added the general metabolism differences in the part and discussion.
Especially, for the general metabolism patways, DEGs were mostly enriched in including the starch and sucrose, pyrimidine, purine, and galactose under control (Figure 4A).
Under control condition, multiple DEGs were detected, which most involved in the metabolism pathways such as the starch and sucrose, pyrimidine, purine, and galactose. Take the starch and sucrose metabolism for example, previous studies reported that this metabolism pathway played important role in the responses refulation under various stresses such as salt (Oliveira et al., 2013), water (Chang and Ryan, 1987; Liao et al., 2016), and drought (Netrphan et al., 2012). Our study identified that the chieh-qua cultivars with heat-reisitant differences exerted significantly differential expression on starch and sucrose metabolism, indicating this DEGs might contribute to their resistant differences on heat when exposed to high temperture condition.
Line 239: heat-tolerant
Response: We have changed it to “heat-tolerant”.
Line 240: heat-sensitive
Response: We have changed it to “heat sensitive”.
Line 247: what is normal leaf?
Response: In our opinion, we thought the normal leaf referred to leaves without external damage. However, we thought that the word “normal”should be deleted in the part.
Lines 251-254: it is much easy to use polysaccharide stating for leaf structure.
Response: Thank you for your advice. We would use this method for further research.
Line 276: please, clarify how much replications.
Response: It’s 9 replications in this assay with three technical and biological replicates, respectively.
Figure 6 legend/layout does not fit with the text. Please, clarify TF name on it.
Similar comments to other figures.
Response: We are sorry for that the numbers of Figures don’t correspond with the text. We have clarified them in the legends of Figure 5-7.
Reviewer 3 Report
The manuscript falls within the general scope of the journal. However, the paper can be reconsidered after major revisions.
The novelty of the paper should be clearly indicated (Introduction).
The physiological state can change along one simple leaf. Did the author use the same leaf part and the leaves of the same sizes for measurements?
Indicate the PPFD to which the plants were exposed. What about air humidity?
References confirming tolerance/sensitivity of inbred lines should be provided in M&M
SOD measurement was not described in M&M.
The discussion needs to also include the limitations of this study and further experiments to answer important questions resulting from this study. Are there any implications for future research?
Author Response
Dear reviewer
Thank you very much for handling our manuscript carefully entitled “Leaf transcriptome analysis exploring of candidate genes for heat stress responses in chieh-qua (BenincasahispidaCogn. var. Chieh-qua How)” (Manuscript ID: 442467). We have carefully considered your comments, and revised the manuscript so that all concerns were taken into account.
The manuscript falls within the general scope of the journal. However, the paper can be reconsidered after major revisions.
Response: Thank you very much for your advice. We would carefully edit the manuscript according to your suggestions.
The novelty of the paper should be clearly indicated (Introduction).
Response: We have added the novelty of the paper in the Introduction part. And the last part was changed to “In this study, we obtained two differently heat resistant chieh-qua cultivas and firsly carried out RNA-sequencing analysis to explore the transcriptional variations under normal and high temperture conditions, respectively. Different stress-responsive novel transcript isoforms were identified and we also analyzed the differential gene expression patterns in response to heat stresses. Functional categorization of differentially expressed transcripts was carried out to reveal various metabolic pathways involved in heat stress responses. Overall, this study, as the first transcriptome-related study in chieh-qua, not only provides a theoretical basis for further study of the regulatory mechanism of heat tolerance, but also enriches the chieh-qua genome database, which would be helpful in the isolation of crucial genes of chieh-qua.”
The physiological state can change along one simple leaf. Did the author use the same leaf part and the leaves of the same sizes for measurements?
Response: Yes, when we carried out the physiological measurements, we used the same leaf part and the leaves of the same sizein our study.
Indicate the PPFD to which the plants were exposed. What about air humidity?
Response: All A39 and H5 plants were exposed to the PPFD condition. And the air humidity is 60% RH (relative humidity).
References confirming tolerance/sensitivity of inbred lines should be provided in M&M
Response: Thank you very much for your suggestion. We are sorry that we would not provide the related reference, which confirmed their tolerance/sensitivity of inbred lines. As in this study, we firsly identified their heat reisitance or sensitivity of A39 and H5, respectively.
SOD measurement was not described in M&M.
Response: We have added the SOD measurement in the Materials and Method.
SOD activity of A39 and H5 was detected by NBT method (nitroblue tetrazolium) (Giannopolitis and Ries, 1977; Paranidharan et al., 2005). In detail, the chieh-qua leaves (1 g) were homogenized in liquid nitrogen and mortar in 10 ml of 0.2 M lysate (citrate phosphate buffer (pH 6.5) with 0.5% Triton X-100). Then, the mixture was centrifuged at 10,000 g for 30 min at 4℃ and the supernatant was the enzyme source. The riboflavin was added last after the enzyme extract was finished and A560 was measured for their OD values.
[1] Giannopolitis CN, Ries SK. 1977. Superoxide dismutase. Plant Physiol 59:309 – 314
[2] Paranidharan V, Palaniswami A, Vidhyasekaran P, et al. A host-specific toxin of Rhizoctonia solani triggers superoxide dismutase (SOD) activity in rice. Archiv für Pflanzenschutz, 2005, 38(2):151-156.
The discussion needs to also include the limitations of this study and further experiments to answer important questions resulting from this study. Are there any implications for future research?
Response: Thank you very much for your suggestions. We have added the limitations of this study and further study plan in the Discussion part.
Therefore, next research should focus on the functional regulation of crucial genes involved in heat response combing physiology, genetics, and molecular biology technology. Especially, the gene’s role on heat stress should be identified by transgentic study.
Round 2
Reviewer 3 Report
The manuscript falls within the general scope of the journal. However, the paper can be accepted after major revisions.
Major remarks:
The title should be written in a way more related and specific to the obtained results or to the most interesting result.
The novelty of the paper should be clearly indicated (Introduction).
Introduction is not concluded by a clear hypothesis, which will be tested and discussed.
Could you clearly indicate what biological and technical replicates are in you research? What constituted a replicate. Were measurements based on one leaf, one group of leaves?
The physiological state can change along one simple leaf. Did the author use the same leaf part and the leaves of the same sizes for measurements?
The discussion needs to also include further experiments to answer important questions resulting from this study. Are there any implications for future research?
Author Response
The title should be written in a way more related and specific to the obtained results or to the most interesting result.
Response: Thank you very much for your suggestion. We have changed our title to “Transcriptome analyses provide novel insights into heat stress responses in chieh-qua (Benincasa hispida Cogn. var. Chieh-qua How)”.
The novelty of the paper should be clearly indicated (Introduction).
Response: The novelty of this paper is that “We firstly carried out RNA-sequencing analysis to explore the transcriptional variations for heat responses in chieh-qua” and “we found several DEGs involved in heat-response.” This part is clearly presented in the last section of Introduction.
Introduction is not concluded by a clear hypothesis, which will be tested and discussed.
Response: We have changed the Introduction to “In this study, we obtained two differently heat resistant chieh-qua cultivas and firstly carried out RNA-sequencing analysis to explore the transcriptional variations under normal and high temperature conditions, respectively. Different stress-responsive novel transcript isoforms were identified and genes related to heat shock proteins (HSPs), ubiquitin-protein ligase, transcriptional factors, and pentatricopeptide repeat-containing proteins were significantly changed after heat stress. Functional categorization of differentially expressed transcripts was carried out to reveal various metabolic pathways involved in heat stress responses. This study provides a theoretical basis in the regulatory mechanism on heat tolerance in chieh-qua.”
Could you clearly indicate what biological and technical replicates are in you research? What constituted a replicate. Were measurements based on one leaf, one group of leaves?
Response: We clearly indicate the “Before treatment and on the 4th day of heat stress, leaf tissue of 10 plants from each pot was sampled and pooled together from normal plants and heat stressed plants of each cultivar, respectively. Three biological replicates were applied for each cultivar. Each biological replicate contained 10 plants randomly.” And measurements were based on a group of leaves.
The physiological state can change along one simple leaf. Did the author use the same leaf part and the leaves of the same sizes for measurements?
Response: In view of the changes along one simple leaf, we used the same leaf part and the leaves of the same sizes for management and investigation.
The discussion needs to also include further experiments to answer important questions resulting from this study. Are there any implications for future research?
Response: In all, the RNA-Seq between different chieh-qua cultivars was firstly carried out to analyze the regulatory mechanism under high temperature. Several crucial genes related to heat shock proteins, ubiquitin-protein ligase, and transcriptional factors were significantly changed important during heat stress. This work not only provided a basis for further understanding the molecular mechanism on heat tolerance, but also exerted valuable and useful genes involved in heat stress, which would be helpful for the genetic improvement of heat tolerant in chieh-qua breeding. Further research should focus on the functional regulation characterization of crucial genes involved in heat response combing physiology, genetics, and molecular biology technology. Most importantly, crucial genes’ roles on heat stress should be identified by transgentic study and other methods.